# Antimicrobial Residues in Food from Animal Origin—A Review of the Literature Focusing on Products Collected in Stores and Markets Worldwide

**DOI:** 10.3390/antibiotics10050534

**Published:** 2021-05-06

**Authors:** Fritz Michael Treiber, Heide Beranek-Knauer

**Affiliations:** Institute of Molecular Biosciences, University of Graz, Humboldtstrasse 50/EG, A-8010 Graz, Austria; heide.knauer@uni-graz.at

**Keywords:** antimicrobial residues, livestock, milk, eggs, poultry, pigs, cattle, seafood

## Abstract

The extensive use of antibiotics leads to antibiotic residues in frequently consumed foods. Generally, the main use of antibiotics in animals is to treat and prevent diseases and growth promotion. However, the residues and their breakdown products have several side effects on the human body and, in a broader sense, on the environment. In relation to the human body, the frequency of mutations is increased, the bone marrow is damaged (chloramphenicol), and the reproductive organs of humans are affected. Carcinogenic effects have been found with antibiotics such as sulfamethazine, oxytetracycline, and furazolidone. We summarized data from 73 scientific studies reporting antimicrobial residues in animal products that were freely available for sale. The studies were published in English starting from 1999 till 2021 and identified through the Pubmed search engine. The aims were to find out which antibiotics, legal or illegal, could be found in animal foods worldwide. Which are stable to get into the food chain and exceed the maximum residue limits (MRL) regarding the EU guidelines as a comparison. Reducing antimicrobial residues in food from animal origin and, in addition to this, fighting the tremendous growth and spread of antimicrobial resistance will undoubtedly be one of the most difficult food safety challenges in the coming years.

## 1. Introduction

The discovery of antibiotics and their use in animals and humans was indisputably one of the greatest achievements of the 20th century [1]. However, the first antibiotic-resistant germs appeared as early as the 1950s [2]. With the generous use of antibiotics in animal breeding, which were also used as growth accelerators, furiously resistant germs emerged and spread [3,4]. In order to be able to cope with this problem, bans have been introduced which have banned the use of human antibiotics in animal husbandry. However, the problems with multi-resistant germs are by no means gone [5]. In some industrialized countries, there are already bans on the use of antibiotics on a preventive basis in animal breeding [6]. These may no longer be used as growth accelerators. Denmark has a prominent role in this context. In 1995, a systematic monitoring program was introduced in this country regarding the use of antibiotics in animal breeding but also in humans. It is known as DANMAP, which means Danish Integrated Antimicrobial Resistance Monitoring and Research Program [7,8]. A reduction in the use of antibiotics could be achieved with this model in Denmark [9]. Implementation of the Danish model is hardly conceivable in developing and emerging countries because there is largely a lack of laws, monitoring bodies, and structures for documentation. The great worldwide demand for animal products, also as a source of protein in the diet, often leads to the uncontrolled use of antibiotics in factory farming [10]. Meat production accounts for 73% of global antibiotic use. Forecasts on antibiotic consumption in relation to the manufacture of animal products anticipate growth of 11.5% by 2030. The sudden accumulation of antibiotic resistance in livestock is especially a big challenge in developing countries. While meat consumption is falling somewhat in industrialized countries, especially in Europe, it is exploding in some developing countries. There, access to veterinary antibiotics is primarily unregulated, and the knowledge of overdosing and the development of resistance among users is deficient. It is clear that the addition of antibiotics to animal feed is the main factor behind the increased incidence and spread of antibiotic resistance [11]. Apart from this severe problem, the disposal of animal dung and the application of antibiotic-contaminated wastewater into the environment are further challenges that are often reflected in the scientific literature. In this context, antibiotic residues in foods of animal origin are also treated relatively rarely, which is another problem area. This was the decisive reason for us to make a review on this topic and the scientific articles already available on it. As for the consumption of antibiotics, the most common prescriptions for people are penicillins, macrolides, and fluoroquinolones. In contrast, tetracyclines and sulfonamides are most commonly used in animal breeding. The residues and degradation products of these substances have many undesirable side effects on the human body. It negatively influences the immune system, damages the kidneys (gentamicin), increases the frequency of mutations, can damage the liver, damages the bone marrow (chloramphenicol), and affects the human reproductive organs. Carcinogenic effects have been found with antibiotics such as sulfamethazine, oxytetracycline, and furazolidone [12]. Maximum residue limits of antibiotics regarding food help protect consumers but are no guarantees that animal products exceeding the limits will not come onto the market and be ultimately consumed by people. Even if the maximum residue limits for antibiotics in foods of animal origin are not exceeded, they can still lead to problems in the long term. For example, antibiotics used in veterinary medicine were found in food, drinking water, and urine of preschoolers in Hong Kong [13]. However, there are also special substances that need a closer look. Nitrofurazone is an antibiotic with carcinogenic properties. It can affect the DNA of cells and result in genetic toxicity that can possibly result in cancer [14]. This antibiotic has been found time and again, especially in shrimps but also in aquaculture catfish [15]. A particular challenge, in this case, is that this molecule is not so easy to detect. The right technology and the associated equipment are required so that reliable evidence can be provided [16]. Especially in underdeveloped countries regarding food safety, this is undoubtedly why nitrofurazone often goes undetected, and thus, contaminated products are also distributed on the world market and ultimately consumed.

Our review aims were as follows: (a) Which antibiotics were detected in foods of animal origin and which test methods were used? (b) How often do consumers get in contact with food contaminated with antibiotics? What country differences are there based on the published studies regarding food animal species and antimicrobial residues? (c) How often have the MRLs (EU guidelines) been exceeded in the case of positive evidence? We extracted all raw data and metrics reported in these studies, discussed the limitations of the methodologies used, and documented data gaps. We hope that this review will help regarding that. The detection of antibiotic residues in animal foods is increasingly investigated and documented, and thus, more data is made available to the scientific community.

## 2. Results

The number of first search results was huge. So, the summaries of the scientific texts had to be sorted out by hand based on the question in order to include as many papers as possible. Antibiotic residues in animal products that reached consumers had to be neatly separated from antibiotic degradation tests in tissues and methods for detecting these compounds. This resulted in the data after the first review, which was already much smaller as can be seen in Table 1. By choosing the search terms, some works were found twice. These were removed again in the last step. The 83 papers were considered as a whole and evaluated regarding eight data points. In total, 73 papers contained information regarding all eight data points.

As can be seen in Figure 1, the number of publications increased steadily. Between 2016 and 2020, the number almost doubled in relation to the previously evaluated period.

For a clearer evaluation, all studies with salmon, other fish from aquaculture, and shrimp were combined as seafood in the evaluation, as can be seen in Figure 2. Cattle [17,18,19,20,21,22], Eggs [23,24,25,26,27,28,29,30], Milk [31,32,33,34,35,36,37,38,39,40,41,42,43,44,45,46,47,48,49,50,51,52,53,54,55,56,57], Pork [58,59,60], Poultry [58,59,60,61,62,63,64,65,66,67], Sheep [68] and seafood [58,69,70,71,72,73,74,75,76,77,78,79,80,81,82,83]. Combined analysis of more than one animal product was very common for pork, poultry, and seafood [63,84,85,86,87,88,89,90,91,92]. The data from the combined papers were divided among the individual groups of animal products. The milk from cows was an abundant source for the detection of antibiotics. This was followed by seafood, poultry, and cattle. Sheep and lamb were seldom examined.

In Vietnam, China, and Iran, most of the studies related to antibiotic residues in animal products have been carried out, as can be seen in Figure 3. Vietnam, as well as China, produce animal products for the world market. In Iran, on the other hand, products are manufactured for the domestic market. Thailand, which, like Vietnam, produces a lot of seafood for the international market, is quite far behind in terms of the number of studies. In China, the increasingly strict legal requirements are apparently taking effect. A government program also set out to massively reduce the use of antibiotics in agriculture. However, it should be noted that the small number of studies from China regarding the number of animal products produced in that country raises further questions.

Compared to Asia, the picture in Europe is entirely different, as can be seen in Figure 4. The studies are almost evenly distributed across different countries. This could be caused by the effects of the strict directives in EU policy on antibiotic residues in foods of animal origin. In many European Union countries, antibiotic residues in animal products are checked by state agencies and published nationally in annual reports.

In the data from Africa, as can be seen in Figure 5, Nigeria stands out in particular. Much of the research on the African continent comes from the most populous country in West Africa, followed by Tanzania and other Eastern African countries. North Africa is only represented with Egypt, Algeria, and Morocco with regard to investigations into antibiotic residues. The most frequently contaminated products in this evaluation group were milk, eggs, and beef. As raw or pasteurized, milk was a rich source of antibiotic residues above and below the permitted limit values.

The situation on the American continent is entirely different, as can be seen in Figure 6. Here is the agar country Brazil with seven works at the top. The USA, a large agar country on this continent, has only one study may be due to a strict policy regarding the use and final concentration of antibiotics in animal products as in the EU.

The determination of antibiotic residues and the determination of their concentration in animal products were carried out in the 73 studies using different test procedures. Table 2 shows all methods of use.

In the evaluated scientific studies, as can be seen in Table 2, the high-performance liquid chromatography (HPLC) method was used most frequently. We also added the high-performance liquid chromatography coupled with UV/VIS and UHPLC-MS/MS methods under this section. As far as laboratory analysis methods are concerned, the liquid chromatography-tandem mass spectrometry followed with some distance. Ready-made test kits were often used in order to be able to detect a range of antibiotic residues easily. The following tests were used: The Charm Blue Yellow antibiotic residue test kit (Charm Sciences Incorporated, MA, USA), Premi^®^test technology (R-Biopharm, Darmstadt, Germany), Copan test kit (Christian Hansen Company, Hoersholm, Denmark), Delvo SP^®^ test kit (SP mini kit; Delft, The Netherlands), Equinox test, SNAP tetracycline test kit, SNAPbeta-lactam test kit, the SNAPgentamicin test kit (Idexx Laboratories, Westbrook, ME, USA), Ridascreen chloramphenicol, and streptomycin kit (R-Biopharm, Darmstadt, Germany),. Microbial Inhibitor Tests and test kits were used, such as the microbial inhibition test using Bacillus subtilis ATCC-6633 (Thermo Fisher Diagnostics Limited, Warrington, UK).

The use of ready-made test kits was prevalent in African countries. This may also be due to the fact that other laboratory facilities were not available. In some cases, the test kits were used to detect antibiotic residues in the samples initially and then subject them to a more detailed examination method or determine the residues more precisely using this method.

Considering all 73 scientific papers evaluated, tetracycline was detected in 39 cases, as can be seen in Table 3. This antibiotic was found most often in animal products in the scientific papers that we analyzed. Most of the records came from Africa (15), followed by Asia (13), both Americas (10), and one record from Europe. The second most common antibiotic detected 22 times is oxytetracycline. Again, the most records for this substance came from Africa (10), followed by Asia (7), America (4), and Europe with one record. Penicillin G was detected in 16 papers. Penicillin G is known to cause allergies in some people and is, therefore, a danger in animal foods for the people concerned and should not be underestimated. The discovery of chloramphenicol, which should no longer be present in animal products, is also very worrying. This was proven in 12 studies. Regarding the records of this substance, Africa leads again with (6) cases followed by Asia (4), Europe (2), and America (0). Especially with the two works that were done in Europe, namely in Slovenia and Turkey, it is worth taking a closer look at the studies. From a total of 1308 random milk samples collected from 1991 to 2000, only one sample was contaminated with chloramphenicol at a concentration of 4.6 µg kg(−1) [54]. In a relatively recent study from turkey, chloramphenicol residues in sea bream (*Sparus aurata*) and sea bass (*Dicentrarchus labrax*) were found in 18.3% of the samples (*n* = 82) [69]. Scientific papers from Turkey were assigned to Europe in our evaluation. However, Turkey is not a member of the European Union and therefore also has different legislation or other directives or statutory inspection bodies regarding the use of antibiotics in agriculture. Two other substances, such as nitrofurantoin (proven in four studies) and furazolidone (proven in one study), are also on the list of prohibited substances that have no place in food.

The maximum residue limit (MRL) is the maximum allowed concentration of residue in a food product obtained from an animal that has received a veterinary medicine or that has been exposed to an antibiotic for use in animal husbandry. The substances found include a wide range of antibiotics used in agriculture. The most critical question in this context, however, is how many substances exceeded the upper limit values. This is displayed in Figure 7. In 52 cases, antibiotic residues were found that exceeded the limit values. Out of the total number of samples taken, only less than 5% of animal foods were affected.

## 3. Discussion

In this work, we reviewed 73 studies on antimicrobial residues in animal products published in English since 1999. In relation to the importance of this topic, the number of publications found was relatively small. This finding can have several causes. In the USA, the European Union, and some Asian countries, the legal guidelines regarding antibiotic residues in food are relatively strict and are also checked by state authorities. A scientific study by non-governmental research groups, therefore, does not seem worthwhile. Unfortunately, there are often rumors among the population in industrialized countries that the foods available there are full of antibiotics, pesticides, and hormones, which would cause serious health problems. It must be noted here that it is precisely in these countries that food safety is highest in terms of residues of various chemicals and pathogenic germs or parasites. Many European countries have already banned the addition of antibiotics to animal feed as a preventive measure in relation to the avoidance of diseases in livestock farms [6,93]. Additionally, the European Union has introduced the maximum limit values for residues of various substances in milk. However, it must be said here that such limit values do not guarantee the absence of drug residues in milk or dairy products. In addition, the study situation is deplorable if one looks at the spread of antibiotic residues from milk in processed dairy products. How high the number of residues is and how they affect the human body has hardly been researched [94]. The situation is different in many emerging and developing countries in Asia, Africa, and South America. Many factors promote the presence of antibiotic residues in meat products, especially in developing countries. The main reasons are as follows: (1) The time each antibiotic needs to be broken down or excreted in the animal organism often lacks for both breeders and butchers; (2) there is no complete monitoring from the prescription to the use of the substances; (3) the detection methods are often inadequate or not available at all, in order to comply with limit values; and (4) the absence of certification systems regarding food products from animal origin. A wide range of antibiotics has been found in the following products: milk, eggs, poultry, beef, pork, seafood, fish, and mutton. Particular attention should be paid to antibiotics that are toxic to humans even in low concentrations, such as chloramphenicol and tetracycline. Various studies have shown that antibiotic residues from food can negatively impact human health in the form of allergic reactions, mutations in cells, the development of imbalances in the intestinal microbiome, and, ultimately, multi-resistant germs. In addition to this, rapid tests, which include many of the antibiotics used in animal breeding, have been the only means of analysis in some countries. Such evidence is somewhat vague but at least allows conclusions to be drawn as to which antibiotics can be found in food or are increasingly used on farms. The same applies to the few papers that used microbiological assays.

Products that exceed the maximum allowed residue limits pose a serious problem. Heat treatments that occur in cooking processes can reduce the risk of ingestion of sulfonamides, tetracyclines [95,96], and fluoroquinolones but do not guarantee their breakdown or the complete breakdown of these antibiotics’ residues in animal products, such as broiler meat. The high stability of quinolones [97] and β-lactams [98] represents a significant risk to human health because the residues of these antibiotics can remain in milk after heat treatment and, therefore, can reach the dairy industry and consumers. Regarding white shrimp (*Penaeus indicus*) that contained chloramphenicol residues, this drug gets destroyed or degraded during heat processing treatments [99]. Furthermore, the toxic effects of the degradation products will also have to be examined more closely in the future. Their formation in cooked meat should definitely be included in the maximum limits for antibiotic residues when these are set.

Apart from that, concentrations of antibiotic residues vary between different edible muscle tissues [100]. However, in the case of muscle tissue, the regulatory process does not differentiate between different edible muscle types in poultry. Previous studies showed that higher fluoroquinolone residue concentrations in breast versus thigh muscle [101].

The previous findings of this work were able to locate most of the problems with antibiotic residues in developing countries and emerging economies. It must not be forgotten that due to world trade, many of these animal products also end up on the plates of western consumers. Seafood and fish from aquaculture should be mentioned here. Due to the many positive properties in relation to a balanced diet, the consumption of these animal products is also increasing sharply [102,103]. This means that antibiotic residues are also imported, and incorrect information on origin is an increasing problem. Monitoring these imports as completely as possible would also be advisable in western countries [71]. Efficient detection methods are also available in these countries.

In many European countries, as well as in the USA, there is a trend in animal breeding and husbandry to keep them in their natural habitat, away from factory farming in large stables. In addition, consumers want more and more meat from animals that have been treated with as few antibiotics as possible during rearing and until slaughter. However, can this desire of consumers be implemented without any problems? In a Danish study regarding consumer preferences for meat with reduced antibiotic use in pig production, two-thirds of the consumers stated a positive will to pay more for this product. However, only one in five were willing to pay more than 10% for pork produced using antibiotics, 20% less than average [104]. While consumers are increasingly demanding pork products that have been reared without antibiotics, bacteria and parasites multiply precisely among these pigs, which are subsequently associated with human diseases that are ingested through food or arise through this route. A comparative study on pigs that were freely bred with antibiotics and others that came from conventional breeding showed clear results. The pigs that were bred outdoors under natural conditions had noticeably higher rates of questionable germs than the pigs from conventional breeding. The incidence of Salmonella was significantly higher in the antibiotic-free-bred herds than in the control herd. Therefore, it is questionable to what extent pigs that have been bred freely with antibiotics are considered safe food [105].

The presence of antibiotic residues in meat products that are above the maximum permitted limits is a serious problem. Based on the values of the maximum levels of residues in animal tissue and in relation to the daily consumption calculated on the bodyweight of the person, these residues should not pose a health risk. Exceptions are allergic reactions to antibiotics. In this context, however, a possible connection must be mentioned.

There appears to be a relationship between antibiotic use and the increase in obesity in people. The number of publications on this topic is increasing steadily. Some authors point out that there is a direct link between the increased use of antibiotics in animal breeding and the increase in the number of obese people in our society over the past 70 years [106]. It would be possible that the long-term and low-dose intake of antibiotic residues via animal food products has permanently disrupted the balance of the microbiome in the intestine, which was subsequently jointly responsible for the weight gain [107,108]. The intestinal microbiome has many tasks that only became known or could be researched in more detail in the last 15 years. The microorganisms help with the digestion of food and with the release of various nutrients, keep various pathogenic germs in check, stimulate or modulate the immune system and have tasks in communication regarding the intestinal-brain axis [109]. Furthermore, there are indications that an imbalance in the intestinal microbiome at a young age can lead to numerous metabolic disorders in later years of life. These include diabetes, various allergies or intolerances, obesity, complaints in the cardiovascular system, and neurological problems.

## 4. Materials and Methods

### 4.1. Scientific Paper Selection

The PubMed engine (National Library of Medicine-8600 Rockville Pike Bethesda, MD 20894 https://pubmed.ncbi.nlm.nih.gov/, the website was accessed on 15 March 2021) was used to search for original scientific papers published in English from January 1999 to March 2021. The following terms were used to search scientific papers using the following keywords: (antimicrobial* OR antibiotic*) AND (cattle* OR lamb* OR sheep* OR poultry* OR egg* OR pork* OR pig* OR livestock* OR shrimp* OR salmon* OR milk* OR seafood*). All retrieved records were saved for further review. Publications not reporting original research data or written in languages other than English were further excluded. Publications containing antimicrobial residues data in the abstract were selected for a first investigation. All scientific papers that dealt with residues of antibiotics were excluded, but in the sense of detection methods or residues in the environment, as well as pollution of animal manure, the rest that contained only research data regarding antimicrobial residues in animal food were reviewed with full content. In addition to the techniques used to detect antibiotic residues, the size of the sample was also an important quality criterion for us. We divided the sample sizes into 6 categories. Up to 50 samples were documented in 16 publications. Up to 100 samples were documented in 19 papers. Up to 200 samples were documented in 20 papers. Up to 300 samples were documented in 12 papers. Up to 400 papers were documented in 6 papers. Twelve papers were summarized for a sample size of over 400 up to 25,000. All work with a sample number below 50 was subjected to a thorough analysis regarding sampling and analytical methods. If this was correct or meaningful, this data was included in our evaluation. Publications were broadly classified by the country where the research took place, year of publishing, and further categorization regarding the animal food product.

### 4.2. Data Extraction

From each selected publication, the following information was compiled as separate records (data points): (1) country of study, (2) year, (3) type of animal product, (4) sample size, (5) animal production type, (6) antibiotics found, (7) detection method, and (8) over maximum residue limits.

All data were entered as single records (‘datapoints’) in Excel (Microsoft Office). Antimicrobials and antimicrobial classes listed were those included in the World Organisation for Animal Health (OIE) classification: veterinary critically important antimicrobial (VCIA) agents (10 classes); veterinary highly important antimicrobial (VHIA) agents (8 classes); and important veterinary antimicrobial (VIA) agents (8 classes) [17].

## 5. Conclusions

The problem of antimicrobial residues in animal products is not new. However, due to the globalization of the food trade, we are constantly facing new challenges. Reduction of the antimicrobial volumes currently used by humans and livestock and limiting the use of broad-spectrum and critically important antimicrobials would be essential. Educational work regarding antimicrobial residues in animal products and the associated development of multi-resistant germs must be stepped up in developing countries. This can only be reduced through increased monitoring of the use of antibiotics in animal breeding. In any case, users must be informed about the health consequences for their customers regarding antibiotic residues in animal products and the development of multi-resistant germs. The ban on the disposal of rubbish, sewage, and manure containing antibiotics must also be mentioned in this context. Alternatives to antibiotics, such as various vaccinations, the use of phages, and phage therapy, must be promoted. The use of prebiotics and probiotics in animal feed and the use of traditional medicinal herbs must also be further researched to ultimately reduce the use of antibiotics in animal breeding to a realistic amount.

## Figures and Tables

**Figure 1 antibiotics-10-00534-f001:**
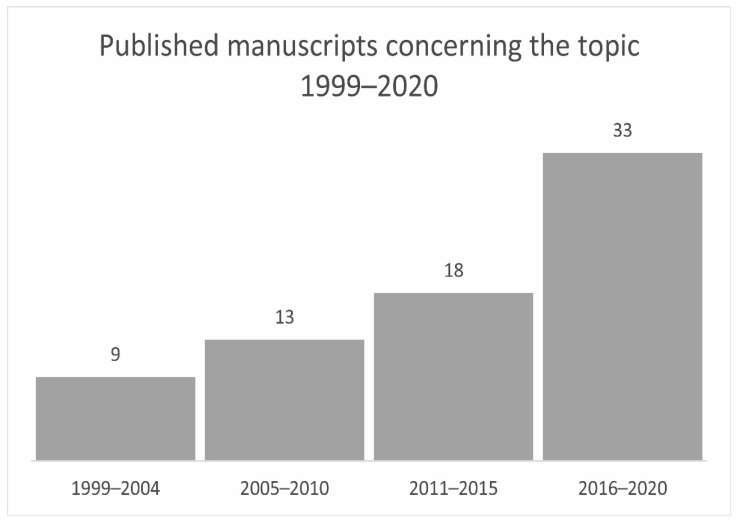
Evaluation of the data in relation to the publication years. For the sake of clarity, the years have been grouped into four groups.

**Figure 2 antibiotics-10-00534-f002:**
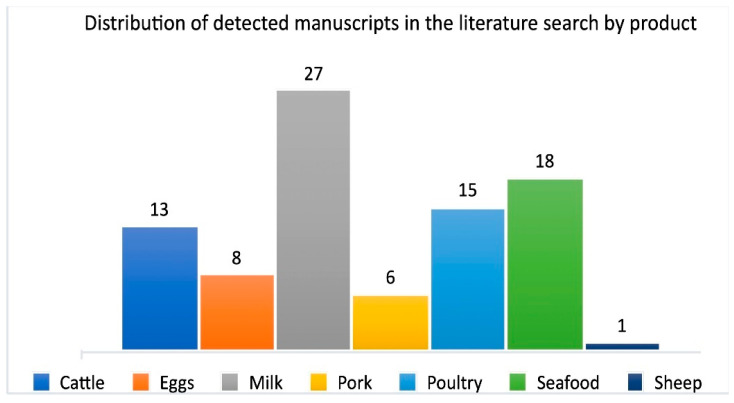
Distribution of manuscripts describing animal products where antibiotics still were detectable.

**Figure 3 antibiotics-10-00534-f003:**
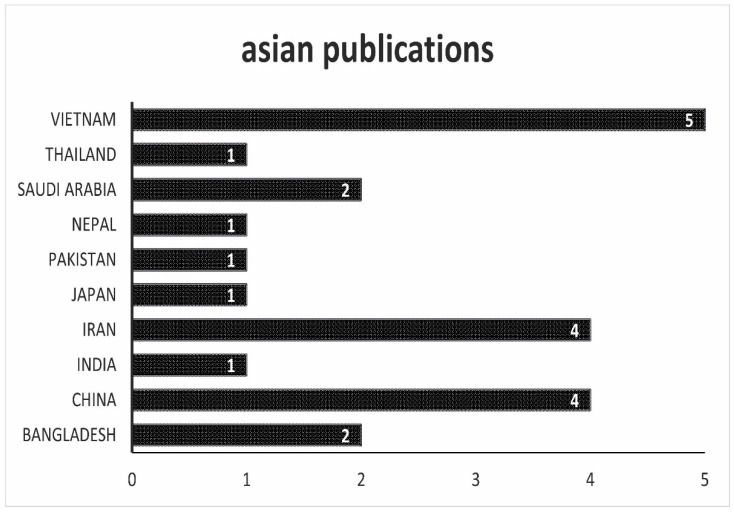
Overview of the origin of the scientific work. In this figure, all works that come from Asia are listed.

**Figure 4 antibiotics-10-00534-f004:**
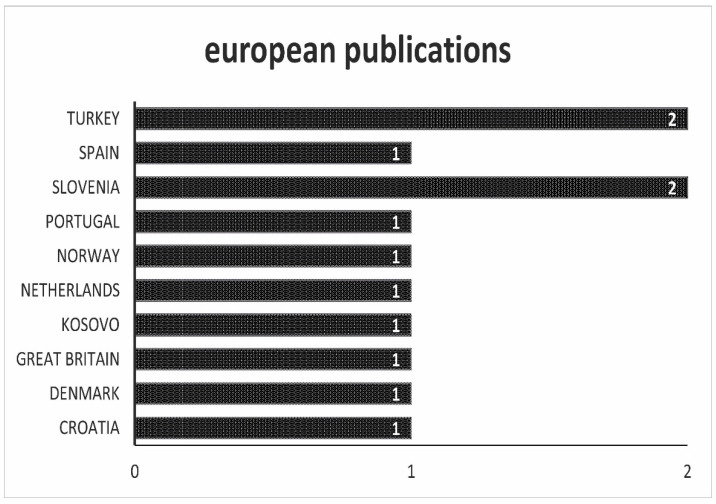
Overview of the origin of the scientific work. In this figure, all works that come from Europe are listed.

**Figure 5 antibiotics-10-00534-f005:**
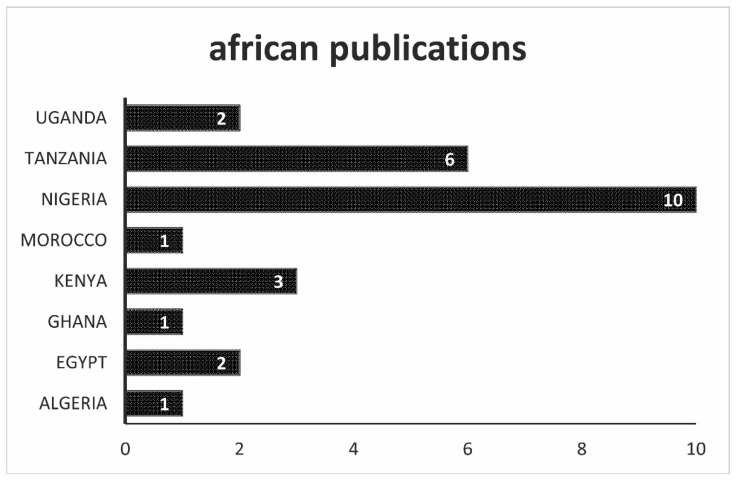
Overview of the origin of the scientific work. In this figure, all works that come from Africa are listed.

**Figure 6 antibiotics-10-00534-f006:**
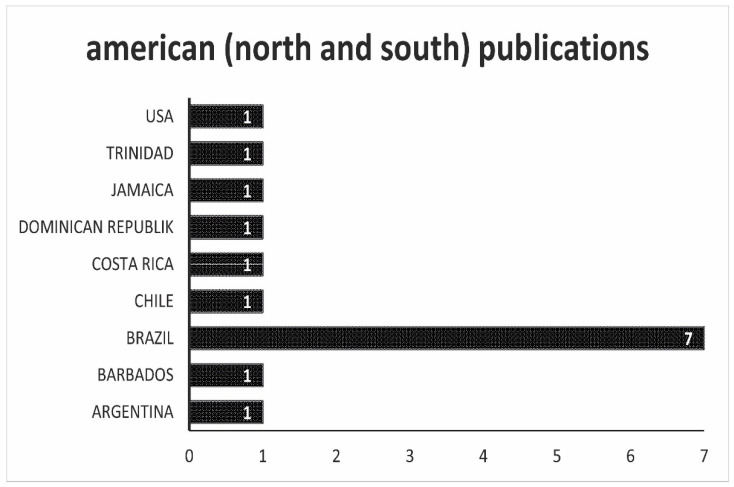
Overview of the origin of the scientific work. In this figure, all works that come from both American continents are listed.

**Figure 7 antibiotics-10-00534-f007:**
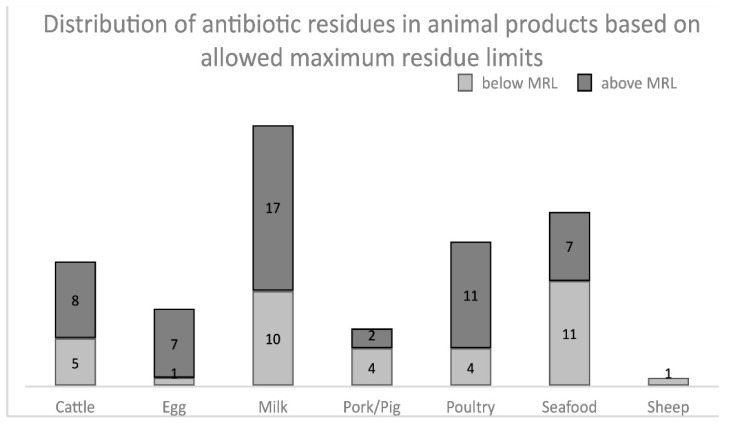
Distribution of antibiotic residues in animal products based on allowed maximum residue limits (MRL) regarding the 73 reviewed papers.

**Table 1 antibiotics-10-00534-t001:** Summary of the search results displayed in three steps.

Keyword Search	Primary Search Results	Remaining after Abstract Review	Remaining after Removal of Duplicates
*Antimicrobial residues* AND *livestock*	381	16	8
*Antimicrobial residues* AND *seafood*	202	16	11
*Antimicrobial residues* AND *shrimp*	177	5	2
*Antimicrobial residues* AND *cattle*	1469	15	5
*Antimicrobial residues* AND *poultry*	1272	19	10
*Antimicrobial residues* AND *pork*	91	8	0
*Antimicrobial residues* AND *pig*	877	15	4
*Antimicrobial residues* AND *salmon*	95	5	0
*Antimicrobial residues* AND *milk*	1148	32	24
*Antimicrobial residues* AND *egg*	767	8	8
*Antimicrobial residues* AND *lamb*	274	1	0
*Antimicrobial residues* AND *sheep*	12	1	1
Total	6765	141	73

**Table 2 antibiotics-10-00534-t002:** Methods and techniques used to detect various antibiotic residues in animal products.

	ELISA	U/HPLC	GC	LC-MS	Inhibitor Test	Test Kits
Cattle	4	5	0	3	0	1
Egg	1	3	0	0	0	4
Milk	4	6	1	4	1	12
Pork/Pig	1	3	0	2	0	0
Poultry	2	5	0	3	4	1
Seafood	2	7	0	8	0	2
Sheep	0	0	0	0	0	1
Total	14	29	1	20	5	21

**Table 3 antibiotics-10-00534-t003:** Antimicrobial residues found in different animal products.

Antibiotic Detected	Cattle	Egg	Milk	Pork/Pig	Poultry	Seafood	Sheep	Total
Amikacin	0	0	0	1	0	0	0	1
Amoxicilin	0	0	6	3	3	0	0	12
Ampicilin	0	0	3	1	2	0	0	6
Aparamycin	0	0	0	0	0	0	0	0
Ceftiofur	0	0	2	0	0	0	0	2
Cephalexin	0	0	2	0	0	0	0	2
Cephalothin	0	0	2	0	0	0	0	2
Cephapirin	0	0	2	0	0	0	0	2
Chloramphenicol	0	4	4	1	0	3	0	12
Chlortetracycline	1	1	1	0	0	0	0	3
Ciprofloxacin	1	0	0	0	3	0	0	4
Clarithromycin	0	0	0	0	0	1	0	1
Clindamycin	0	0	0	1	0	0	0	1
Cloxacillin	0	0	1	0	0	0	0	1
Doxycyclin	0	1	0	1	1	1	0	4
Enrofloxacin	0	0	0	0	3	4	0	7
Erythromycin	0	0	1	1	0	0	0	2
Florfenicol	0	0	0	0	0	1	0	1
Flumequine	0	1	0	0	0	0	0	1
Fluoroquinolon	1	0	0	1	2	2	0	6
Furazolidone	0	0	0	0	0	1	0	1
Gentamycin	0	0	4	1	1	1	0	7
Higromycin	0	0	0	1	0	0	0	1
Kanamycin	0	0	0	1	0	0	0	1
Lincomycin	0	0	0	1	0	0	0	1
Neomycin	0	0	1	1	1	0	0	3
Nitrofurantoin	0	2	0	1	0	1	0	4
Norfloxacin	0	0	0	0	1	0	0	1
Oxytetracycline	5	3	5	1	5	3	0	22
Penicilin G	7	0	9	0	0	0	0	16
Quinolone	1	1	1	0	2	0	0	5
Roxithromycin	0	0	0	0	0	1	0	1
Streptomycin	1	0	3	1	1	0	0	6
Sulfanilamide	0	1	0	0	0	0	0	1
Sulfadiazine	1	1	3	3	0	2	0	10
Sulfaguanidine	0	0	0	0	0	1	0	1
Sulfamethazine	1	1	2	1	4	1	0	10
Sulfisoxazole	0	0	1	0	0	0	0	1
Sulfmethoxazole	1	0	0	0	0	4	0	5
Sulfmethoxydiazine	1	1	2	0	0	2	0	6
Sulfmethoxypyridazine	0	1	1	0	0	0	0	2
Sulfaquinoxaline	1	0	0	0	2	0	0	3
Sulphanilamide	0	0	0	0	0	0	0	0
Tetracycline	8	3	13	3	8	3	1	39
Thiamphenicol	0	0	0	0	0	1	0	1
Tobramycin	0	0	0	1	0	0	0	1
Trimethoprim	0	0	0	0	1	2	0	3
Trimetoprim	0	0	0	0	0	0	0	0
Tylmicosin	0	0	0	1	0	0	0	1

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
