# Peer review of "Antimicrobial Residues in Food from Animal Origin—A Review of the Literature Focusing on Products Collected in Stores and Markets Worldwide"

_antibiotics, 2021, doi:10.3390/antibiotics10050534_

Round 1

Reviewer 1 Report

The review paper presents a valuable contribution in the study of antimicrobial residues in food from animal origin. The authors collected the recent papers focusing on this topic worldwide and compared, summarized and discussed the findings published in those papers plus made their own conclusions. When published, the paper can be considered a primary source of information about current situation in this area.

My only request for revision concerns the formal layout.

The authors should be more careful with formatting their paper  (e.g. to avoid unnecessary use of colours in the text, lack of gaps or presence of extra gaps, sentence with no verbs).

Author Response

Comment #1:

My only request for revision concerns the formal layout.

The authors should be more careful with formatting their paper (e.g. to avoid unnecessary use of colors in the text, lack of gaps or presence of extra gaps, sentence with no verbs).

Author response:

Dear reviewer, thank you very much for the feedback on our scientific article. As far as the formatting of the text is concerned, we have once again revised it intensively. We also asked a colleague whose mother tongue is English to look it over again for any errors. We have also incorporated the suggestions of the other reviewers into the final draft of our paper. You should receive this as soon as possible. Thank you very much.

Reviewer 2 Report

You need to specify the type of review the manuscript falls under. is this a narrative review, systematic review, or non-systematic review? There is no place under the materials and methods where the quality of the articles included in this review was mentioned. Please add one or two sentences on the quality of articles used for the revirew. Page 6 or 20: Note that Nigeria is a West African country and not located in central Africa. (YOu can find that section just under figure 5) Please correct that in the manuscript. Table 3: Can you please describe how geographical location e.g. continents differed in terms of antimicrobial residues detected? e.g from 39 cases of tetracycline residues in meat products, 10 were reported from Africa ... etc. This provides the context in terms of which country/region/continents and what antimicrobial residue is prevalent.

Author Response

Comment #1:

You need to specify the type of review the manuscript falls under. is this a narrative review, systematic review, or non-systematic review?

Author response:

We have moved our work to a different category, from review to systematic review!

Comment #2:

There is no place under the materials and methods where the quality of the articles included in this review was mentioned. Please add one or two sentences on the quality of articles used for the review.

Author response:

In terms of quality, we have linked the scientific work to the sample size. We have divided the sample sizes into 6 categories. Up to 50 samples were documented in 16 publications. Up to 100 samples were documented in 19 papers. Up to 200 samples were documented in 20 papers. Up to 300 samples were documented in 12 papers. Up to 400 papers were documented in 6 papers. 12 papers were summarized for a sample size over 400 up to 25,000. All work with a sample number below 50 was subjected to a thorough analysis regarding sampling and analytical methods. If this was correct or meaningful, this data was included in our evaluation.

Comment #3:

Page 6 or 20: Note that Nigeria is a West African country and not located in central Africa. (You can find that section just under figure 5) Please correct that in the manuscript.

Author response:

In the data from Africa as can be seen in Figure 5, Nigeria stands out in particular. Much of the research on the African continent comes from the most populous country in West Africa. Followed by Tanzania and other Eastern African countries.

Comment #4:

Table 3: Can you please describe how geographical location e.g. continents differed in terms of antimicrobial residues detected? e.g from 39 cases of tetracycline residues in meat products, 10 were reported from Africa ... etc. This provides the context in terms of which country/region/continents and what antimicrobial residue is prevalent.

Author response:

Based on your constructive feedback, we have incorporated further data from our evaluation into the article, in particular the geographical distribution of the detection of tetracycline, oxytetracycline and chloramphenicol in animal foods. This antibiotic was found most often in animal products in the scientific papers that we analyzed. Most of the records came from Africa (15) followed by Asia (13), both America (10) and one record in Europe. The second most common antibiotic detected 22 times is oxytetracycline. Again, the most records for this substance came from Africa (10) followed by Asia (7), America (4) and Europe with one record. Penicillin G which was detected in 16 papers.

Regarding the records of this substance, Africa leads again with (6) cases followed by Asia (4), Europe (2) and America (0). Especially with the two works that were done in Europe, namely in Slovenia and Turkey, it is worth taking a closer look at the studies. From a total of 1308 random milk samples collected from 1991 to 2000, only one sample was contaminated with chloramphenicol at a concentration of 4.6 microg kg(-1).[54] In a relatively re-cent study from turkey, chloramphenicol residues in sea bream (Sparus aurata) and sea bass (Dicentrarchus labrax) were found in 18.3% of the samples (n=82).[69]. Scientific papers from Turkey were assigned to Europe in our evaluation. However, Turkey is not a member of the European Union and therefore also has different legislation or other directives or statutory inspection bodies regarding the use of antibiotics in agriculture.

Reviewer 3 Report

The paper titled “Antimicrobial residues in food from animal origin – A review of the literature focusing on products freely available in stores and markets” is an interesting article that provides new and important insights about antimicrobial residues in food from animal origin and their connection with human cancers. This is an unpopular topic for this paper should be perfectly written. As a reviewer I suggest WITHDRAWN this article and prepared them according to “Instructions for the Authors” and the article needs English correction. Some minor comments in PDF.

Author Response

Comment #1:

The paper titled “Antimicrobial residues in food from animal origin – A review of the literature focusing on products freely available in stores and markets” is an interesting article that provides new and important insights about antimicrobial residues in food from animal origin and their connection with human cancers. This is an unpopular topic for this paper should be perfectly written.

Author response:

Due to your constructive feedback, we have changed the title of our work: “Antimicrobial residues in food from animal origin – A review of the literature focusing on products collected in stores and markets worldwide.”

Comment #2:

As a reviewer I suggest WITHDRAWN this article and prepared them according to “Instructions for the Authors” and the article needs English correction.

Author response:

As far as the formatting of the text is concerned, we have once again revised it intensively. We have now used the template from the journal for this purpose. We also asked a colleague whose native language is English to look it over again for any errors. After consultation with the editor of this issue, we have designated our work as systematic review.

Comment #3:

Some minor comments in PDF. Should this be same like in the abstract.

Author response:

We adapted the abstract at the beginning of our work. We have changed the textual repetition in the abstract regarding the introduction this way: The aims were to find out which antibiotics, legal or illegal could be found in animal foods worldwide. Which are stable to get into the food chain and exceed the maximum residue limits (MRL) regarding the EU guidelines as a comparison.

Round 2

Reviewer 3 Report

Dear Authors

References are still not prepared according to instruction ;)

Line 127 Figure 3 The title  Capital letter “Asian

Line 142: this same “European”

Line 152: “African”

Line 164: "American"

All the best and stay safe